# Integrating Language Models into Direct Speech Translation: An Inference-Time Solution to Control Gender Inflection

**Dennis Fucci,**[1,2] **Marco Gaido,**[2] **Sara Papi,**[1,2]
**Mauro Cettolo**,[2] **Matteo Negri**,[2] **Luisa Bentivogli**[2]
[1]University of Trento
[2]Fondazione Bruno Kessler
{dfucci,mgaido,spapi,cettolo,negri,bentivo}@fbk.eu

## Abstract

When translating words referring to the speaker, speech translation (ST) systems should not resort to default masculine generics nor rely on potentially misleading vocal traits. Rather, they should assign gender according to the speakers' preference. The existing solutions to do so, though effective, are hardly feasible in practice as they involve dedicated model re-training on gender-labeled ST data. To overcome these limitations, we propose the first inference-time solution to control speaker-related gender inflections in ST. Our approach partially replaces the (biased) *internal* language model (LM) implicitly learned by the ST decoder with gender-specific *external* LMs. Experiments on en→es/fr/it show that our solution outperforms the base models and the best training-time mitigation strategy by up to 31.0 and 1.6 points in gender accuracy, respectively, for feminine forms. The gains are even larger (up to 32.0 and 3.4) in the challenging condition where speakers' vocal traits conflict with their gender.[1]

## 1  Introduction

The problem of gender bias in automatic translation particularly emerges when translating from genderless or notional gender languages (e.g., English) – which feature limited gender-specific marking – into grammatical gender languages (e.g., Spanish) – which exhibit a rich lexical and morpho-syntactic system of gender (Savoldi et al., 2021). In this scenario, when gender-neutral words are translated into gender-marked words (e.g. en: *the nurse* – es: *el/la enfermero/a*), both machine translation (MT) and speech translation (ST) systems are often biased towards masculine or stereotypical predictions (Cho et al., 2019; Prates et al., 2020; Bentivogli et al., 2020; Costa-jussà et al., 2022), especially

in absence of explicit cues (en: *the nurse and his dog*). A common instance is represented by words that refer to the first-person subject (henceforth referred to as speaker-dependent words, such as *I'm a young nurse*). In these cases, direct ST systems (Bérard et al., 2018) have been shown to rely on vocal traits to determine gender inflections (Bentivogli et al., 2020). This, however, does not eliminate the bias toward masculine forms and is not inclusive for those individuals whose vocal properties do not align with their gender, such as people with vocal impairments, children, and transgenders (Matar et al., 2016; Menezes et al., 2022). Therefore, whenever the speaker's gender[1] is known (e.g. in talks or lectures), such information should be exploited to control gender translation and avoid relying on potentially misleading physical cues.

So far, this topic has been investigated only by Gaido et al. (2020). Their best solution consists in creating two gender-specific *specialized* models by fine-tuning a generic direct ST system on sentences uttered by female/male speakers. Though effective, this method has inherent limitations. First, it requires parallel audio-text data labeled with speakers' gender, which are scarcely available and costly to collect. Second, the fine-tunings are computationally demanding as they involve processing audio data, which are much longer (∼8×) than their textual equivalents (Salesky et al., 2019).

To overcome these limitations, we propose the first inference-time solution in direct ST to control gender translation for speaker-dependent words when the speaker's gender is known.[2] Our approach guides gender translation by partially substituting the biased *internal language model* implicitly learned by the ST decoder of a base model with a gender-specific *external language model* learned on monolingual textual data. Through experiments on three language pairs (en→es/fr/it), we demon-

---

[1]Note that, throughout the paper, when using the terms *female, male,* and *gender* we do not refer to speakers' gender identity but exclusively to their preferred linguistic expression of gender (see §8 for an in-depth discussion of this issue).

[2]Code and models available at https://github.com/hlt-mt/FBK-fairseq under Apache License 2.0.

strate that, in terms of gender accuracy, our solution outperforms the base system by up to 31.0 points (for feminine forms) and is on par with the best training-time approach (with up to 1.6 of gain for feminine forms). Its effectiveness is also confirmed when speakers' vocal traits conflict with their gender, with gains up to 32.0 and 3.4 over the base system and the best training-time solution.

## 2 ILM/ELM for Gender Translation

The autoregressive decoder of an encoder-decoder architecture is trained to predict the next target token given the previous ones and the encoder output. Thereby, it implicitly learns to model the target language from the training data, thus developing an *internal language model* (ILM) (McDermott et al., 2019; Variani et al., 2020). We assume that, in a direct ST model trained on unbalanced data where female speakers (and consequently feminine speaker-dependent words) are under-represented (Tatman, 2017), the ILM is biased toward masculine forms. Therefore, we propose to guide the generation of the ST model with respect to speaker-dependent words by substituting the biased ILM with a gender-specific *external language model* (ELM). To this aim, we train two ELMs on monolingual text corpora (easy to collect, unlike labelled audio data) containing either feminine or masculine speaker-dependent words (see §3). At inference time, when we have prior knowledge of the speaker's gender from the metadata, we *i)* integrate the ELM specialized in either masculine or feminine forms (depending on the speaker's gender) into the ST model, and *ii)* (partially) remove the ILM contribution.

The integration of end-to-end models with ELMs is a widespread solution to leverage text data in speech recognition (Bahdanau et al., 2016; Chorowski and Jaitly, 2017; Kannan et al., 2018; Irie et al., 2019). Successful applications span from recognizing rare words (Sainath et al., 2021; Huang et al., 2022) to coping with out-of-vocabulary terms (Hori et al., 2017), domain adaptation (Sriram et al., 2018; Shan et al., 2019) and under-resourced conditions (McDermott et al., 2019). However, to the best of our knowledge, ELM integration has not been explored in the field of direct ST, nor in the context of gender translation, as we do here. Among the various methods proposed for the ELM integration (Gülçehre et al., 2015; Gülçehre et al., 2017; Sriram et al., 2018; Stahlberg et al., 2018;

Shan et al., 2019; McDermott et al., 2019), we avoid those that require training-time interventions, and we resort to *shallow fusion* (Gülçehre et al., 2015; Gülçehre et al., 2017), an effective technique (Kannan et al., 2018; Inaguma et al., 2019) that consists in the log-linear combination of the posterior of the base model ($p_{M_B}$) and the prior of the ELM ($p_{ELM}$).

As regards the ILM removal, which previous studies already shown to amplify the performance gains yielded by ELM integration (Meng et al., 2021a,b,c; Andrés-Ferrer et al., 2021; Liu et al., 2022; Meng et al., 2023), the most critical aspect is its estimation. In fact, since the ILM is implicitly modeled in the decoder, disentangling its contribution from the rest of the network is a challenging task (Variani et al., 2020; Meng et al., 2021a,b; Zeineldeen et al., 2021). Among the estimation methods demonstrated by Zeineldeen et al. (2021) to yield the best results, we select the *global encoder average*, as it does not require training-time interventions. This method computes the ILM prior ($p_{ILM}$) as:

$$p_{ILM}(y) = p_{M_{B_{decoder}}}(y|c)$$

namely, by feeding the ST decoder with the average $c$ of the encoder outputs $h_{n,t}$ over all the $T_n$ timesteps of the $N$ training samples, where $c$ is:

$$c = \frac{1}{\sum_{n=1}^{N} T_n} \sum_{n=1}^{N} \sum_{t=1}^{T_n} h_{n,t}$$

Therefore, given an audio input $x$, the output $\widehat{y}$ of our solution is the translation $y$ that maximizes the log-linear combination of $p_{M_B}$, $p_{ELM}$ and $p_{ILM}$:

$$\widehat{y} = \operatorname*{argmax}_{y} \{ \log p_{M_B}(y|x) - \beta_{ILM} \log p_{ILM}(y) + \beta_{ELM} \log p_{ELM}(y) \}$$

where $\beta_{ILM}$ and $\beta_{ELM}$ are positive scalar weights calibrating ELM integration and ILM removal.

The three components ($p_{M_B}$, $p_{ELM}$, and $p_{ILM}$) convey different information: *i)* $p_{M_B}$ embeds both the acoustic and the linguistic information learned from the ST data; *ii)* $p_{ILM}$ represents the estimated linguistic knowledge learned by $M_B$; *iii)* $p_{ELM}$ embeds linguistic information (in our case gender-specific forms) learned from external textual resources. Therefore, $\beta_{ILM}$ and $\beta_{ELM}$ must be set to values that effectively integrate the internal and external linguistic knowledge, so that the gender

| | es | | | | fr | | | | it | | | |
|---|---|---|---|---|---|---|---|---|---|---|---|---|
| | train | | dev | | train | | dev | | train | | dev | |
| | M | F | M | F | M | F | M | F | M | F | M | F |
| Sent. | 196.8K | 111.9K | 1.6K | 1.2K | 566.9K | 232.4K | 8.5K | 3.3K | 370.7K | 171.9K | 5.3K | 3.0K |
| Words | 4.1M | 2.4M | 37.5K | 26.7K | 13.7M | 5.5M | 232.2K | 87.1K | 8.9M | 4.2M | 132.3K | 75.4K |

Table 1: Statistics for the monolingual text corpora collected.

bias affecting the ST decoder is mitigated by the ELM. At the same time, the linguistic contribution supplied by the ELM must not override the acoustic modelling capabilities of $p_{M_B}$, so as to avoid translation quality drops. Accordingly, we estimate $\beta_{ILM}$ and $\beta_{ELM}$ by optimizing the harmonic mean of the two metrics (gender accuracy and BLEU – see §3) used to measure gender bias and overall translation quality, so as to equally weigh our two objectives. In Appendix A, we discuss the computation of $\beta_{ILM}$ and $\beta_{ELM}$ values, also showing that their precise estimation is not critical since final results are rather robust to small weight variations.

## 3 Data and Metrics

Our en→es/fr/it ST systems are trained on the TED-based MuST-C corpus (Cattoni et al., 2021). This resource includes a manual annotation of the speakers' gender (Gaido et al., 2020), which is used to determine the gender translation of speaker-dependent words. To train the ELMs, we collected GenderCrawl,[3] a set of monolingual corpora for each target language and gender. Each corpus is made of sentences with speaker-dependent words that clarify the speaker's gender (e.g., es: *Soy nueva* <F> *en esta zona* [en: *I am new to this area*], es: *Debía ser fiel a mi mismo* <M> [en: *I had to be true to myself*]). These sentences were automatically selected from ParaCrawl (Bañón et al., 2020) through regular expressions representative of morpho-syntactic patterns matching references to the first-person singular. Additionally, we have also collected a validation set by applying the same regular expressions to the MuST-C training sets. The statistics of all these datasets are presented in Table 1.

We evaluate our systems on the TED-derived and gender-sensitive MuST-SHE benchmark (Bentivogli et al., 2020). In particular, we focus on its "Category 1", which contains from 560 to 607 sentences (depending on the target language) with speaker-dependent words annotated in the refer-

ence. To assess gender translation, we use the official MuST-SHE evaluation script[4], which produces two measures: *i) term coverage*, i.e. the percentage of annotated words that are generated by the system (disregarding their gender marking), and on which gender translation is hence automatically measurable, and *ii) gender accuracy*, i.e. the percentage of words generated in the correct gender among the measurable ones. Lastly, overall translation quality is calculated with SacreBLEU (Post, 2018).[5]

## 4 Results

For each language pair, we evaluate our approach by training: *i)* an ST baseline model ($M_B$) that is not aware of the speaker's gender; *ii)* the specialized models ($M_{SP}$) presented in (Gaido et al., 2020), re-implemented as upper bound to compare our inference-time solution with the best training-time approach; *iii)* the combination of $M_B$ with the gender-specific ELMs and the ILM removal ($M_{B\text{-}ILM+ELM}$); *iv)* a variant of the approach, where the ILM is not removed ($M_{B+ELM}$), serving as an ablation study to disentangle the ILM and ELM contributions. Detailed experimental settings and model description are provided in Appendix B.

### 4.1 Main Results

Table 2 presents BLEU, term coverage, and gender accuracy scores for all language pairs, divided into feminine/masculine (F/M) forms.

**Gender Accuracy.** The results indicate that our approach, both with and without the ILM removal, significantly outperforms $M_B$ on all language pairs. Specifically, $M_{B\text{-}ILM+ELM}$ is always better than $M_{B+ELM}$, demonstrating that the ILM removal in combination with ELM integration improves debiasing. The accuracy gains of $M_{B\text{-}ILM+ELM}$ over $M_B$ are particularly high on feminine forms, ranging from 25.4 to 31.0. In addition, the accuracy of $M_{B\text{-}ILM+ELM}$ is comparable to that of the training-time approach $M_{SP}$. While $M_{SP}$ is significantly superior only for M in en-it and en-fr, $M_{B\text{-}ILM+ELM}$

[4]https://mt.fbk.eu/must-she/.
[5]case:mixed|eff:no|tok:13a|smooth:exp|version:2.0.0

| Models | en-es | | | | | en-fr | | | | | en-it | | | | |
|---|---|---|---|---|---|---|---|---|---|---|---|---|---|---|---|
| | BLEU | Coverage | | Gender Acc. | | BLEU | Coverage | | Gender Acc. | | BLEU | Coverage | | Gender Acc. | |
| | | M | F | M | F | | M | F | M | F | | M | F | M | F |
| $M_B$ | 34.8 | 65.1 | 67.9 | 71.6 | 45.7 | 29.8 | 51.5 | 55.9 | 72.5 | 52.0 | 26.8 | 51.6 | 50.6 | 77.3 | 49.5 |
| $M_{SP}$ | **35.2** | 64.8 | 66.8 | **85.6** | **76.8** | **29.9** | 52.9 | 55.2 | **92.4** | 78.5 | **27.3** | **52.8** | 49.4 | **92.5** | 73.3 |
| $M_{B+ELM}$ | 33.7[ab] | **67.5**[AB] | 68.1 | 77.9[Ab] | 69.2[Ab] | 29.3 | **54.9**[A] | **57.1** | 81.9[Ab] | 75.8[A] | 27.2 | 51.8 | **54.4**[AB] | 81.2[Ab] | 72.8[A] |
| $M_{B\text{-}ILM+ELM}$ | 34.4[b] | 65.8 | **71.2**[AB] | 82.3[A] | 76.7[A] | 29.8 | 54.4[A] | 56.1 | 84.5[Ab] | **79.2**[A] | 27.2 | 52.3 | 54.1[AB] | 84.9[Ab] | **74.9**[A] |

Table 2: BLEU (↑), (term) coverage (↑), and M/F gender accuracy (Gender Acc., ↑) scores. [A/a] and [B/b] indicate that the improvement (uppercase) or the degradation (lowercase) of our technique over the baseline ($M_B$) and the fine-tuning approach ($M_{SP}$), respectively, is statistically significant (bootstrap resampling with 95% CI, Koehn 2004).

is the best on average for F, the most misgendered category.

**Translation Quality.** Looking at BLEU scores, we notice that, with the only exception of en-it, the simple integration of the ELM ($M_{B+ELM}$) degrades the quality with respect to both $M_B$ and $M_{SP}$,[6] especially in en-es where the drops are statistically significant. The ILM removal mostly solves the problem, as $M_{B\text{-}ILM+ELM}$ achieves scores that are comparable to $M_{SP}$ on en-fr and en-it, and partly closes the gap on en-es, where the drop with respect to $M_B$ (-0.4) is not statistically significant. Interestingly, looking at term coverage, both $M_{B\text{-}ILM+ELM}$ and $M_{B+ELM}$ consistently outperform $M_B$ and $M_{SP}$, with the only exception of masculine words in en-it. In particular, the gains are high for feminine words, where $M_{B\text{-}ILM+ELM}$ significantly outperforms both $M_B$ and $M_{SP}$. This shows that the integration of textual data can increase the ability to model feminine vocabulary, less represented in training data.

In conclusion, our inference-time solution effectively improves gender translation in direct ST, especially for feminine forms (see Appendix C for output examples). Moreover, it achieves comparable results with the best training-time approach, while overcoming its limitations. Such improvements do not come at the detriment of the overall translation quality (as shown by BLEU scores) nor of the accuracy in assigning gender to words that pertain to human referents other than the speaker (as shown in Appendix D).

### 4.2 Robustness to Vocal Traits

We also evaluate the inclusivity of our solution for speakers whose vocal traits are stereotypically associated with a gender opposite to their own. As MuST-SHE solely contains utterances from speakers whose gender aligns with their vocal prop-

erties, we simulate this condition using the provided "wrong references", in which the speaker-dependent words are swapped to the opposite gender. We treat them as correct references, so as to have female voices with masculine targets and vice versa, and we require the systems to produce the output with the gender of the target. Table 3 shows BLEU, term coverage, and gender accuracy for $M_B$, $M_{SP}$, and our best-performing model $M_{B\text{-}ILM+ELM}$, averaged over the three language pairs.

**Gender Accuracy.** Regarding gender realization, $M_{B\text{-}ILM+ELM}$ performs noticeably better than $M_B$, as we observe a substantial improvement of 19.7 points in producing masculine forms (Voice F–Gdr M) and 32.0 in producing feminine forms (Voice M–Gdr F). This suggests that our approach is capable of partially overriding the vocal information, on which the base model unduly relies to translate the speaker-dependent words. In comparison with $M_{SP}$, our approach is inferior in Voice F–Gdr M, while it is superior in generating the less-represented feminine translation (Voice M–Gdr F), confirming the trends observed in the previous scenario (see §4.1).

**Translation Quality.** In terms of BLEU, our approach ($M_{B\text{-}ILM+ELM}$) is on par with the training-time strategy ($M_{SP}$), but they both suffer a ~2.5 BLEU drop with respect to the base system ($M_B$). The reason for this drop may lay on the fact that gender-specific models learned patterns that differentiate male and female language (Mulac et al., 2001; Boulis and Ostendorf, 2005), which are disregarded when only swapping the gendered words in the references. However, $M_{B\text{-}ILM+ELM}$ outperforms $M_B$ and $M_{SP}$ in terms of coverage, with a marginal gain (0.5-0.6) for male speakers (Voice M–Gdr F) and a larger gain (2.3-3.7) for female speakers (Voice F–Gdr M), confirming that our approach increases the coverage of the vocabulary used by females (even when expressed in the masculine form).

---

[6] In Gaido et al. (2020), the *specialized* systems achieve higher results as their base models are built using large ST, ASR, and MT corpora, while we train only on MuST-C.

| Models | Average | | | | |
|---|---|---|---|---|---|
| | **BLEU** | **Coverage** | | **Gender Acc.** | |
| | | Voice F Gdr M | Voice M Gdr F | Voice F Gdr M | Voice M Gdr F |
| $M_B$ | **30.5** | 58.0 | 56.2 | 50.9 | 26.1 |
| $M_{SP}$ | 28.2 | 56.6 | 56.1 | **83.0** | 54.7 |
| $M_{B-ILM+ELM}$ | 28.0 | **60.3** | **56.7** | 70.6 | **58.1** |

Table 3: BLEU, term coverage, and gender accuracy for the conflicting scenario averaged over en→es/fr/it.

All in all, the experiments in this challenging testing condition prove that our solution effectively overrides the reliance of base ST systems on speakers' vocal traits. Also, they confirm its superiority in translating the less-represented feminine forms.

## 5 Conclusions

We proposed the first inference-time solution to control gender translation of speaker-dependent words in direct ST. Our approach partially replaces the biased ILM of the ST decoder with a gender-specific ELM. As such, it can be applied to existing models without the need for labeled ST data or computationally expensive re-trainings, overcoming the limitations of existing training-time methods. Experiments on three language pairs proved the effectiveness of our technique in controlling gender inflections of words referring to the first-person subject, regardless of whether the speakers' vocal traits are aligned with their gender or not. In addition to significantly increasing the gender accuracy of base ST models, it achieves substantial parity with the best training-time method while consistently increasing the correct generation of feminine forms.

## 6 Acknowledgements

This work is part of the project "Bias Mitigation and Gender Neutralization Techniques for Automatic Translation", which is financially supported by an Amazon Research Award AWS AI grant. Moreover, we acknowledge the support of the PNRR project FAIR - Future AI Research (PE00000013), under the NRRP MUR program funded by the NextGenerationEU.

## 7 Limitations

In our experiments, we exclusively evaluated our approach on English to Romance language translations. Conducting experiments on different language pairs would be valuable. However, it is important to note that such endeavors would demand substantial efforts in annotating data, as benchmarks akin to MuST-SHE are currently unavailable for other target languages.

Our inference-time solution, as described in the paper, significantly reduces the computational costs of current approaches by eliminating the need for ST retraining. However, there is an increase in inference costs, due to the additional forward passes on the ELM and ILM (which is the same as the ST decoder, but fed with a different encoder output). In particular, since our implementation has not been optimized and performs the operations sequentially, our solution reduces the inference speed (computed as the number of generated tokens per second) by ∼40% (from 165 to 100).[7] Such slowdown can be reduced by: *i)* parallelizing the forward passes of the ST model, ELM, and ILM; *ii)* caching computed states in the ILM to avoid recomputation at each generation step. Optimizing our implementation, although necessary for production usage, is outside the scope of our work.

Lastly, our ELM implementation uses the same BPE (Sennrich et al., 2016) vocabulary of the ST models, trained on the textual target of MuST-C. Due to the under-representation of feminine forms in this corpus, statistical segmentation methods like BPE split the less frequent feminine forms into less compact sequences of tokens (for example, in our experiments, we observed the split *maes_tra* vs *maestro* for Spanish). This tokenization process can penalize generalization on morphology and, consequently, gender translation when compared to character-level representations (Belinkov et al., 2020). As such, an interesting future direction is represented by training the ELMs with a character-based vocabulary, which has the potential to enhance gender accuracy and further increase the significant gains already achieved.

## 8 Ethics Statement

In this paper we presented a new methodology to improve ST systems in their ability to correctly generate masculine and feminine forms for first-person-singular referents. Hereafter, we contextualize the impact of our research and discuss the ethical principles at the basis of our work.

We define gender bias in MT/ST as the tendency of systems to systematically favor masculine forms to the detriment of the feminine ones

---

[7]Statistics computed on a p3.2xlarge instance on AWS (featuring one NVIDIA V100 GPU).

when related to human entities (Crawford, 2017). This bias not only hampers the performance of the system by producing erroneous translations of gender-marked words, but also has significant societal implications. For example, incorrect gender translations can impact self-perception, as linguistic expressions of gender play a crucial role in negotiating and communicating personal representation (Stahlberg et al., 2007; Corbett, 2013; Gygax et al., 2019). According to Blodgett et al. (2021) and Savoldi et al. (2021), gender bias in translation technologies leads to both *representational harms*, such as under-representation of women and diminished visibility of their linguistic repertoire, and *allocational harms*, characterized by unequal quality of service due to performance disparities between male and female users.

In light of the above, we believe that our solution positively impacts single individuals and society at large, by improving not only the experience of using such technologies but also feminine visibility. Furthermore, by relying on explicit gender information, our mitigation solution goes beyond a mere and potentially misleading exploitation of the speech signal. Indeed, using speaker's vocal properties would foster the stereotypical expectations about how masculine or feminine voices should sound, which is not inclusive for certain users, such as transgender individuals or people with laryngeal diseases (Matar et al., 2016; Pereira et al., 2018; Villas-Bôas et al., 2021; Menezes et al., 2022).

As regards possible concerns about the gender information considered in our experiments, we relied on the annotations of the two datasets used, MuST-C/MuST-Speakers and MuST-SHE. Both these resources have been manually annotated with speakers' gender information based on the personal pronouns found in their public TED profile (Gaido et al., 2020; Bentivogli et al., 2020). We follow the statement of the curators of these resources, thus bearing in mind that the gender tag accounts only for the linguistic gender by which the speakers accept to be referred to in English and to which they would like the translation to conform. We acknowledge that this information does not necessarily correspond to the speakers' self-determined gender identity (Cao and Daumé III, 2020). We are also aware that we cannot consider their preference as static in time (Lauscher et al., 2022).

Last but not least, in this work we only consider binary linguistic forms as they are the only ones represented in the currently available ST data. In fact, to the best of our knowledge, ST corpora also representing non-binary speakers are not yet available. However, we encourage a vision of gender going beyond binarism and we believe that extending the application of our method to non-binary forms (e.g. by integrating a third, *non-binary* ELM) can be an interesting extension of this work.

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

## A  Contributions of $\beta_{ILM}$-$\beta_{ELM}$

As stated in §2, our method relies on two hyper-parameters ($\beta_{ELM}$ and $\beta_{ILM}$). In this section, we report their optimal values (§A.1), and discuss the impact of varying these values on the results (§A.2).

### A.1  Optimal $\beta_{ILM}$-$\beta_{ELM}$ Combinations

In the lack of a validation set with the same characteristics of MuST-SHE, we used this same benchmark for a 10-fold cross validation. At each iteration, we translate the held-out data with the pair $(\beta_{ILM}, \beta_{ELM}) \in \{0.00, 0.05, \dots 0.95, 1.00\}^2$ that maximizes the harmonic mean between gender accuracy and BLEU (see §2) on the validation folds. At the end of this process, the whole MuST-SHE was fairly translated and ready for evaluation, and $\beta_{ILM}$ and $\beta_{ELM}$ were robustly estimated.

However, in a real use case, we need a unique combination of $\beta_{ELM}$ and $\beta_{ILM}$ for each gender. Therefore, in Table 4 we report the mean values of $\beta_{ELM}$ and $\beta_{ILM}$ over the 10 folds for each language pair. We can notice that the optimal values are closely aligned across the three language directions. In general, for $M_{B\text{-}ILM+ELM}$ $\beta_{ELM}$ is always higher than $\beta_{ILM}$. Moreover, another clear and consistent trend emerging in all language pairs is the necessity for higher $\beta_{ELM}$ and $\beta_{ILM}$ values when the speaker is female. In this condition, a higher contribution of the ELM is required to counterbalance the inherent bias of the base ST model towards masculine forms.

### A.2  Impact of $\beta_{ILM}$ and $\beta_{ELM}$

In addition to empirically estimating $\beta_{ILM}$ and $\beta_{ELM}$ through cross-validation, we also investigated the importance of optimizing the balance between the ILM and the ELM for mitigating bias without compromising translation quality. To this end, for each language direction we computed the performance variations by adjusting $\beta_{ILM}$ and $\beta_{ELM}$ in increments of 0.05. Figure 1 shows BLEU and gender accuracy (calculated globally for F and M) scores for each $(\beta_{ILM}, \beta_{ELM})$ combination. Each heatmap defines a space bounded by the base ST model (bottom left corner: $(\beta_{ILM}, \beta_{ELM}) = (0.0, 0.0)$) and by the ST model with the ILM totally replaced by the gender-specific ELMs (top right corner: $(\beta_{ILM}, \beta_{ELM}) = (1.0, 1.0)$).

The trends are similar for all the three language directions. As for gender accuracy, ELM integration appears to be more critical than ILM removal. Specifically, we observe that the accuracy improves as the value of $\beta_{ELM}$ increases. Looking at BLEU, we observe a diagonal ellipse-shaped trend with higher scores around the bottom left corner. This indicates that, to preserve translation quality, $\beta_{ILM}$ and $\beta_{ELM}$ should be similar and not too high. Overall, although the trends for translation quality and gender accuracy differ, the two objectives share high results in the middle area.

Most importantly, we can notice that the results are not significantly affected by small variations in the weights, with wide smooth areas with similar scores and no isolated peaks. This demonstrates the robustness of our solution with respect to a suboptimal estimation of $\beta_{ILM}$ and $\beta_{ELM}$.

## B  ST Model and Language Models

**ST Models**  Our direct ST models are made of a 12-layer Conformer (Gulati et al., 2020) encoder, in light of its favorable results in ST (Inaguma et al., 2021), and a 6-layer Transformer (Vaswani et al., 2017) decoder. The architecture is also preceded by two 1D convolutional layers with 5 as kernel size and stride 2, as per (Wang et al., 2020). We use 512 embedding features, 2,048 hidden features in the FFN, and a kernel size of 31 for Conformer convolutions. In total, the ST models have 116M parameters. We trained them with an auxiliary CTC loss on the 8th encoder layer (Gaido et al., 2022) and we leveraged the CTC module to compress the sequence length (Liu et al., 2020; Gaido et al., 2021). We encoded text into BPE (Sennrich et al., 2016) using SentencePiece (Kudo and Richardson, 2018) with a vocabulary size of 8,000 (Di Gangi et al., 2020), and we used Adam optimizer (Kingma and Ba, 2015) ($\beta_1 = 0.9$, $\beta_2 = 0.98$) and Noam learning rate (lr) scheduler (Vaswani et al., 2017) (inverse square-root) starting from 0 and reaching the 0.002 peak in $25,000$ warm-up steps. The ST

| Models | en-es | | | | en-fr | | | | en-it | | | |
|---|---|---|---|---|---|---|---|---|---|---|---|---|
| | M | | F | | M | | F | | M | | F | |
| | $\beta_{ILM}$ | $\beta_{ELM}$ | $\beta_{ILM}$ | $\beta_{ELM}$ | $\beta_{ILM}$ | $\beta_{ELM}$ | $\beta_{ILM}$ | $\beta_{ELM}$ | $\beta_{ILM}$ | $\beta_{ELM}$ | $\beta_{ILM}$ | $\beta_{ELM}$ |
| $M_{B\text{-}ILM+ELM}$ | 0.200 | 0.250 | 0.285 | 0.390 | 0.155 | 0.245 | 0.215 | 0.355 | 0.125 | 0.310 | 0.195 | 0.305 |
| $M_{B+ELM}$ | - | 0.145 | - | 0.310 | - | 0.235 | - | 0.300 | - | 0.195 | - | 0.275 |

Table 4: Mean of the optimal values for $\beta_{ILM}$ and $\beta_{ELM}$ found using 10-fold cross-validation.

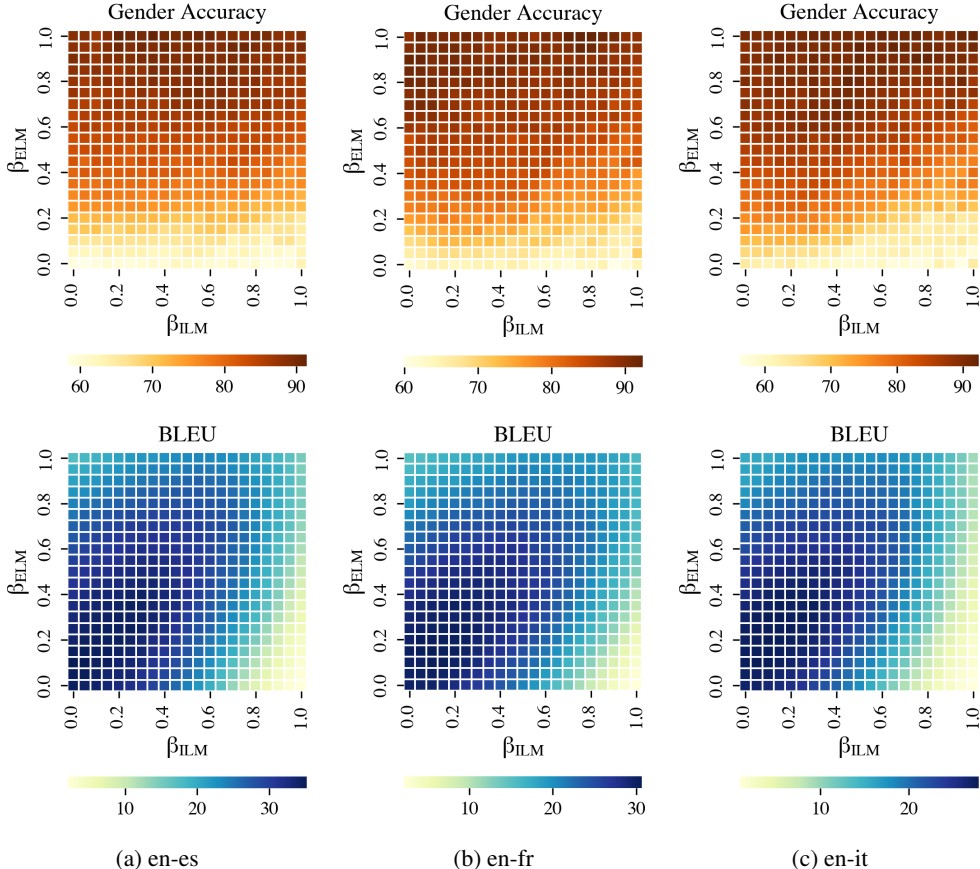

Figure 1: BLEU and gender accuracy heatmaps with different combinations of $\beta_{ILM}$ and $\beta_{ELM}$ for all language pairs.

models for each language direction were trained for 50k steps on 4 NVIDIA A100 GPUs (40GB of RAM) with 40k tokens per mini-batch and 2 as update frequency, and we averaged the last 7 checkpoints. To implement the specialized models ($M_{SP}$), we fine-tuned $M_B$ on the masculine/feminine partitions of the MuST-C data, with a constant lr of 0.001 for 7 epochs, and we averaged the last 4 checkpoints. All our models are implemented on fairseq (Ott et al., 2019).

**Language Models** The gender-specific ELMs are Transformer decoders with 6 layers (23M weights) trained with the same vocabularies and hyper-parameters of $M_B$, except for the learning rate warm-up updates that we set to 200. We early stopped the training after 5 epochs without im-

provements on the validation loss, and we average the 5 checkpoints around the best on the validation set.

## C Examples

In Table 5 we report output samples that well exemplify the behavior of our models and the baseline.

First, the examples in en-fr and en-it confirm the gender-accuracy improvements of our methods discussed in §4.1. The outputs of the baseline ($M_B$) contain speaker-dependent words with the wrong gender, as a masculine form (fr: *fatigué*, en: *tired*) is used with a female speaker in en-fr, and a feminine form (it: *assunta*, en: *hired*) with a male speaker in en-it. Our solution ($M_{B\text{-}ILM+ELM}$), instead, consistently generates the correct gender inflection in both cases (fr: *fatiguée* and it: *as-*

| Lang. | Gender | | Example |
|---|---|---|---|
| en-es | F | SRC | I felt **alienated**, **intimidated** and **judged** by many. |
| | | REF | Me sentí **alienada**, **intimidada** y **juzgada** por muchos. |
| | | $M_B$ | Me sentí **alienada**, *intimidante* (EN. *intimidating*) y **juzgada** por muchos. |
| | | $M_{B\text{-}ILM+ELM}$ | Me sentí **alienada**, **intimidada** y **juzgada** por muchos. |
| | | $M_{B+ELM}$ | Me sentí *aislada* (EN. *isolated*), **intimidada** y **juzgada** por muchos. |
| en-fr | F | SRC | I was **tired** of faking normal. |
| | | REF | J'étais **fatiguée** de simuler la normalité. |
| | | $M_B$ | J'étais **fatigué** d'avoir l'air normal. |
| | | $M_{B\text{-}ILM+ELM}$ | J'étais **fatiguée** d'avoir l'air normal. |
| | | $M_{B+ELM}$ | J'étais **fatigué** d'avoir *l'impression d'être normal* (EN. *of having the impression of being normal*). |
| en-it | M | SRC | In 2007, I was **hired** as a **curator** at the Denver Museum of Nature and Science. |
| | | REF | Nel 2007, fui **assunto** come **curatore** al Denver Museum of Nature and Science. |
| | | $M_B$ | Nel 2007 sono stata **assunta** come **curatore** al Museo d'*Arte Moderna di* Science (EN. *Modern Art of*). |
| | | $M_{B\text{-}ILM+ELM}$ | Nel 2007 sono stato **assunto** come **curatore** al Museo d'*Arte Moderna di* Science (EN. *Modern Art of*). |
| | | $M_{B+ELM}$ | Nel 2007 sono stato **assunto** come **curatore** al Museo d'*Arte Moderna di Scienza* (EN. *of Modern Art of Science*). |

Table 5: Examples of outputs from the baseline $M_B$, $M_{B\text{-}ILM+ELM}$ and $M_{B+ELM}$, along with the corresponding source (SRC) and reference (REF). We indicate the correct/wrong gender translation for **words** on which gender accuracy is evaluated, as well as generic mistranslations of other *words*.

| Models | en-es | | | | en-fr | | | | en-it | | | |
|---|---|---|---|---|---|---|---|---|---|---|---|---|
| | Coverage | | Gender Acc. | | Coverage | | Gender Acc. | | Coverage | | Gender Acc. | |
| | M | F | M | F | M | F | M | F | M | F | M | F |
| $M_B$ | 72.91 | **68.81** | **82.54** | 63.99 | **66.60** | 59.57 | **84.74** | 68.61 | 58.55 | **60.30** | 81.27 | 64.63 |
| $M_{SP}$ | **73.75** | 66.79 | 82.48 | 65.83 | 64.45 | 58.71 | 83.51 | 69.18 | 57.10 | 59.01 | **82.38** | **68.20** |
| $M_{B\text{-}ILM+ELM}$ | 72.41 | **68.81** | 81.40 | **66.59** | 63.09[a] | **59.78** | 82.87 | **69.87** | 57.74 | 56.87[a] | 81.44 | 67.36[A] |

Table 6: (Term) coverage (↑) and M/F gender accuracy (Gender Acc., ↑) scores for Category 2 of MuST-SHE. [A/a] and [B/b] indicate that the improvement (uppercase) or the degradation (lowercase) of our technique over the baseline ($M_B$) and the fine-tuning approach ($M_{SP}$), respectively, is statistically significant (bootstrap resampling with 95% CI, Koehn 2004).

*sunto*), even without the ILM removal ($M_{B+ELM}$). This is in line with the analysis in Appendix A, where we have seen that gender accuracy mostly depends on ELM integration.

Looking at the en-es example, instead, $M_B$ correctly assigns the gender but it wrongly translates one of the adjectives referred to the speaker, using the epicene term *intimidante* (en: *intimidating*) for *intimidated*. Similarly, the output of $M_{B+ELM}$, although with the correct gender, contains an error (*alienated* is rendered as *aislada*, en: *isolated*). Instead, all adjectives are correct in the output of $M_{B\text{-}ILM+ELM}$, confirming its higher coverage (see §4.1) and the importance of ILM removal to avoid quality drops (see Appendix A and the BLEU scores in §4.1). The latter aspect also emerges from the errors introduced by $M_{B+ELM}$ with respect to $M_B$ both in en-fr and in en-it, which are not present in the output of $M_{B\text{-}ILM+ELM}$: for instance, in en-fr, the translation of *faking normal* alters it meaning, deviating to *avoir l'impression d'être normal* (en: *having the impression of being normal*).

# D   Impact on Human Referents Other than the Speaker

Our work is dedicated to the gender translation of speaker-dependent words i.e., those words that refer to the first-person-singular referent. However, the improvements in handling this aspect should not come to the detriment of the accuracy in assigning the gender to referents different from the speaker. To ensure that this is not the case, we also evaluated the gender translation on the "Category 2" of the MuST-SHE benchmark. This contains approximately 500 sentences with the annotation of words related to third-person references, whose gender is independent from that of the speaker. The results are presented in Table 6.

As for gender accuracy, we observe that all systems are close for masculine forms (M), with variations that are not statistically significant. The largest difference amounts to 1.87 points on en-fr between the baseline ($M_B$) and our solution ($M_{B\text{-}ILM+ELM}$). Similarly, $M_{B\text{-}ILM+ELM}$ and the spe-

cialized systems ($M_{SP}$) achieve comparable scores on feminine forms (F) while $M_B$ is constantly worse, with a statistically significant difference in en-it.

Looking at the term coverage, we do not see clear trends across language pairs. For F, $M_{B\text{-}ILM+ELM}$ suffers from a significant drop in en-it with respect to $M_B$ while it achieves the best scores in en-es and en-fr. For M, there is a significant drop in en-fr, which is not confirmed in the other two language pairs. In addition, the differences with $M_{SP}$ are always ascribable to random fluctuations.

All in all, we can conclude that our debiasing solution specifically designed for speaker-dependent words does not significantly alter the gender assignment for referents different from the speaker.