# OpenReview forum: "Integrating Language Models into Direct Speech Translation: An Inference-Time Solution to Control Gender Inflection"
_EMNLP/2023/Conference — EMNLP 2023 Main_

### Official Review · Reviewer_maVW · 2023-07-30

**Soundness:** 3

**Excitement:**

3: Ambivalent: It has merits (e.g., it reports state-of-the-art results, the idea is nice), but there are key weaknesses (e.g., it describes incremental work), and it can significantly benefit from another round of revision. However, I won't object to accepting it if my co-reviewers champion it.

**Paper Topic And Main Contributions:**

The work tries to solve the important task of translating the speaker dependent words (gender) correctly. Usually the speech translation systems (ST) are heavily biased towards the masculine forms, thereby degrading the performance of the system.

The authors propose to use an external language model (ELM) trained on gender-specific corpora. They do this in combination with effectively removing the existing bias in internal language model (ILM) (a model implicitly trained in the decoder part of the E2E ST system). All of this is done at inference time, which means there is no need to retrain existing ST systems for this technique to be effective.

The proposed system is applied to a number of settings: en->it/es/fr. The system seems to outperform the baselines both in terms of coverage of terms, and gender accuracy.

**Questions For The Authors:**

My questions are a repetition from the above. They are:

- [A] How would we choose which ELM to pick (male/female)? Does this require us to know the speaker’s gender beforehand, i.e., at inference time? This seems like a drawback as the accuracy should be calculated after using a gender detection model in the pipeline (at least in the cases where vocal traits match speaker identity).
- [B] What happens when a single audio file has two speakers (male and female) conversing with each other? Which ELM to pick in that case?

**Reasons To Accept:**

- Extremely important task to solve
- To use this method, there is no need to retrain existing ST systems. It is an inference-time solution. Additionally, it is a ubiquitous method -- it can apply to any E2E ST system.
- The improvements using the proposed method are significant.
- Very interesting application of existing work to remove prior bias of ILM.
- Good coverage of hyperparameters.
- Nice ablation study to show that the model does not solely rely on vocal traits.

**Reasons To Reject:**

- The proposed system seems to deter the model in terms of their BLEU scores (system degrades in 2 out of the 3 settings). This leads me to think that while the model seems to do well on speaker specific terms/inflections, the overall translations degrade.
- How would we choose which ELM to pick (male/female)? Does this require us to know the speaker’s gender beforehand, i.e., at inference time? This seems like a drawback as the accuracy should be calculated after using a gender detection model in the pipeline (at least in the cases where vocal traits match speaker identity).
- What happens when a single audio file has two speakers (male and female) conversing with each other? Which ELM to pick in that case?

**Reproducibility:**

4: Could mostly reproduce the results, but there may be some variation because of sample variance or minor variations in their interpretation of the protocol or method.

**Reviewer Confidence:**

4: Quite sure. I tried to check the important points carefully. It's unlikely, though conceivable, that I missed something that should affect my ratings.

---

> ### Author Rebuttal · Authors · 2023-08-29
>
> We thank the reviewer for their comments and questions. Through our responses, we hope to provide compelling explanations  and convince them to revise the very low soundness score that unduly penalizes our work.
>
> > The proposed system seems to deter the model in terms of their BLEU scores (system degrades in 2 out of the 3 settings).
>
> We respectfully disagree with the reviewer's concern about the degradation in BLEU scores. As per lines 251-252 and results in Table 1, the differences in BLEU scores are not statistically significant, neither when our method  ($M_{B-ILM+ELM}$) has a higher score (+0.4 on en-it) than the baseline, nor when the score is lower (-0.4 on en-es). Accordingly, these differences are ascribable to random fluctuations and, as stated in lines 266-268, we reiterate that our experiments demonstrate that our gender-controlling approach does not affect (neither improves, nor degrades) the overall translation quality.
>
> > How would we choose which ELM to pick (male/female)? [...] the accuracy should be calculated after using a gender detection model in the pipeline (at least in the cases where vocal traits match speaker identity).
>
> As stated in lines 46-50 and 415-424, the use of automatic gender detection raises serious ethical concerns, as expressed in multiple authoritative sources, including the EU AI ACT [1], the EMNLP Ethics Review Questions [2], and the EMNLP Ethics FAQ [3]. This is motivated by the fact that it should never be assumed that the speakers’ vocal traits are aligned with their gender identity; *“the cases where vocal traits match speaker identity”* are known only once we know the speaker’s gender, which makes the introduction of a classifier useless. Accordingly, our approach advocates for explicit provision of gender information (see lines 50-54). This requirement is intrinsic to the task and extends beyond our method, being also shared by the alternative training-time gender-controlling approach we compare against.
>
> [1] “[...] bans on intrusive and discriminatory uses of AI systems such as […] Biometric categorisation systems using sensitive characteristics (e.g. gender, [...])”, see https://www.europarl.europa.eu/news/en/press-room/20230505IPR84904/ai-act-a-step-closer-to-the-first-rules-on-artificial-intelligence
>
> [2] “For papers using identity characteristics (e.g. gender, race, ethnicity) as variables: Does the paper use self-identifications (rather than attributing identity characteristics to participants)?”, see https://2021.emnlp.org/call-for-papers/ethics-review-questions
>
> [3] “[...] avoid either attributing (i.e. assigning, without asking) identity characteristics to the people whose language is studied (guessing their gender, race, nationality, etc)”, see https://2021.emnlp.org/call-for-papers/ethics-faq
>
> > What happens when a single audio file has two speakers (male and female) conversing with each other? Which ELM to pick in that case?
>
> In such scenarios, the typical translation pipeline is composed of several steps. Separating segments belonging to multiple speakers conversing with each other is relevant to tasks such as automatic audio segmentation and/or speaker diarization, which fall outside the scope of this (short) paper. Furthermore, the challenges from multi-speaker scenarios are inherent to the overarching task of controlling gender translation and common to all the other approaches. Thus, addressing these challenges is not necessary to demonstrate the soundness of our approach nor to soundly compare it against other gender-controlling methods.

---

### Official Review · Reviewer_jwAu · 2023-08-02

**Soundness:** 3

**Excitement:**

4: Strong: This paper deepens the understanding of some phenomenon or lowers the barriers to an existing research direction.

**Paper Topic And Main Contributions:**

The paper presents an inference time approach to controlling first-person gender inflection in speech translation.   The problem to be addressed is that 'a direct ST model trained on unbalanced data where female speakers (and consequently feminine speaker-
095 dependent words) are under-represented the[internal language model of the translation system] is biased toward masculine forms'.  The inference time approach refers to use of a gender suitable 'external language model' to guide the correct choice of target language gender;  this is in contrast to using 'gender specific specialised models' that are fine-tuned on training data with female/male speakers.     The paper studies English -> Spanish/French/Italian based on the  MuST-C corpus.   A simple modelling technique for incorporating gender appropriate 'external' LMs into decoding is presented;    the technique appears not to be novel itself, but is apparently novel to this task.   Gains in gender accuracy and coverage (as defined in MuST-C) are reported.    The specialised models serve as a performance target;  the inference time approach gives gains relative to the baseline and in several conditions approaches performance of the specialised models.  A challenging `conflicting scenario' is also reported ,  in which the system is made to produce gendered output contrary to the original gender associated with the audio.     There is a also an extensive limitations section and a good ethics section.

**Questions For The Authors:**

At line 39 the variable c introduced, which doesn't appear to be subsequently mentioned.   Is this needed?   Should it appear in p_{M_B} in the equation at line 153?



**Reasons To Accept:**

The paper presents a reasonable amount of work for a short paper.    The problem is interesting,  the proposed solution is simple,   and the results are based on one of the few available relevant data sets.

**Reasons To Reject:**

The paper starts with ambitious statement (in the abstract): `When translating words referring to the speaker, speech translation (ST) systems should not resort to default masculine generics nor rely on potentially misleading vocal traits. Rather, they should assign gender according to the speakers’ preference. '    This is laudable, but its not clear that the MuST-C collections are the best resource for such work.  Section 4.2 on `Robustness to Vocal Traits' ,  while well motivated,  seems to illustrate the problem.    The aim is to investigate `inclusivity of our solution for speakers whose vocal traits are stereotypically associated with a gender opposite to their own.'  but there is the note that `MuST-SHE solely contains utterances from speakers whose gender aligns with their vocal properties' and so the condition needs to be simulated.    There is also the conclusion that 'our approach increases the coverage of the vocabulary used by females (even when expressed in the masculine form).'    The need to tease this apart suggests the underlying data set was not designed with this task in mind.  This is worth emphasizing since, from the modelling point of view,  the proposed incorporation of different language models is straightforward, and so much of the value of the work would depend on the experimental set up and results.

**Reproducibility:**

3: Could reproduce the results with some difficulty. The settings of parameters are underspecified or subjectively determined; the training/evaluation data are not widely available.

**Reviewer Confidence:**

2: Willing to defend my evaluation, but it is fairly likely that I missed some details, didn't understand some central points, or can't be sure about the novelty of the work.

---

> ### Author Rebuttal · Authors · 2023-08-29
>
> We thank the reviewer for their comments and questions, as they offer us the chance to
> elucidate some aspects of our work.
>
> > its not clear that the MuST-C collections are the best resource for such work [...] MuST-SHE solely contains utterances from speakers whose gender aligns with their vocal properties' and so the condition needs to be simulated.
>
> To the best of our knowledge, MuST-C and MuST-SHE represent the best resources to test our approach, due to the following reasons:
> 1. They feature a manually curated annotation of speakers’ gender, an essential element for the task we address (see lines 181-184).
> 2. MuST-SHE is the only test set within the speech domain featuring a section specifically dedicated to speaker-dependent words (see lines 196-210), which enables the evaluation of gender translations for such words. The only other test set dedicated to gender bias in ST – WinoST – lacks words referring to the speakers.
> 3. MuST-SHE is also used in previous work on the topic [1], where the conflicting condition has been simulated as in our Section 4.2 (we will clarify this aspect in the camera ready). Hence, our evaluation aligns with the related studies.
> 4. The absence of voices presenting vocal traits not aligned to gender identity, such as for transgender speakers, is a prevalent limitation across all existing speech datasets. Consequently, simulating conflicting conditions becomes the only possible way to guarantee a comprehensive and fair evaluation.
>
> In light of these unique characteristics and their use in prior research, we believe that the adoption of MuST-C and MuST-SHE is well motivated and we invite the reviewer to reconsider it as a reason to reject our work.
>
> [1] M. Gaido, B. Savoldi, et al.: Breeding gender-aware direct speech translation systems. In Proc. of COLING 2020.
>
> > At line 39 the variable c introduced, which doesn't appear to be subsequently mentioned. Is this needed? Should it appear in p_{M_B} in the equation at line 153?
>
> The variable $c$ is useful for the understanding of the global encoder average technique (described at lines 145-148). This technique represents our approach to determine the ILM contribution ($p_{ILM}$). Therefore, in line 153, $c$ contributes to $p_{ILM}(y)$, not to $p_{M_B}$. In the camera ready we will define $p_{ILM}(y) =  p_{M_B{}_{decoder}}(y|c)$.

---

### Official Review · Reviewer_6X26 · 2023-08-04

**Soundness:** 4

**Excitement:**

4: Strong: This paper deepens the understanding of some phenomenon or lowers the barriers to an existing research direction.

**Paper Topic And Main Contributions:**

The manuscript proposes an inference-time solution to control gender inflections in speech translation systems for speaker-dependent words. The major contribution of the work is that the proposed solution replaces the internal language model typically used in exiting solutions with gender-specific external language models to assign gender according to the speaker's preference. The experimental results presented on en→es/fr/it show significant improvement in gender accuracy, as well as a gain in training computational costs, in comparison to base ST models, by up to 31.0 and 1.6 points , respectively, for feminine forms. The author(s) also reported results for where the proposed method achieved even higher accuracy and computational gains in special and challenging cases where speakers' vocal traits conflict with their gender. The manuscript concludes by discussing the possible limitations as well as the ethical principles at the basis of their work.

**Reasons To Accept:**

1. Clear, concise and easy to comprehend
2. Research problem and hypothesis clearly defined and backed with results.
3. The proposed inference-time solution to control gender inflections in speech translation systems for speaker-dependent words is effective and outperforms the base models and the best training-time mitigation strategy.

**Reasons To Reject:**

Null

**Reproducibility:**

4: Could mostly reproduce the results, but there may be some variation because of sample variance or minor variations in their interpretation of the protocol or method.

**Reviewer Confidence:**

4: Quite sure. I tried to check the important points carefully. It's unlikely, though conceivable, that I missed something that should affect my ratings.

---

> ### Author Rebuttal · Authors · 2023-08-29
>
> We thank the reviewer for the positive feedback.

---

### Meta-Review · Area_Chair_znzx · 2023-09-15

**Recommendation:** 4

**Metareview:**

Authors propose an inference-time solution to control gender inflections in speech translation. Internal language model is replaced by a gender-specific external language model to choose gender (when generating translation) according to speaker's preference. Main result is a significant improvement in gender accuracy  (for reasonable computational cost). Paper is clear and well written, it is a reasonable contribution for a short paper.
However, there were some discussions about the overall translation quality when gender control is activated (small degradation). The paper should also discuss more on how to select the gender-specific external language model. Overall my recommendation is Accept to Main Conference or Findings

---

### Decision · Program_Chairs · 2023-10-07

**Decision:**

Accept-Main

**Comment:**

Authors propose an inference-time solution to control gender inflections in speech translation. Internal language model is replaced by a gender-specific external language model to choose gender (when generating translation) according to speaker's preference. Main result is a significant improvement in gender accuracy  (for reasonable computational cost). Paper is clear and well written, it is a reasonable contribution for a short paper.
However, there were some discussions about the overall translation quality when gender control is activated (small degradation). The paper should also discuss more on how to select the gender-specific external language model. Overall my recommendation is Accept to Main Conference or Findings